# Development of an antibody fragment that stabilizes GPCR/G-protein complexes

Shoji Maeda[1], Antoine Koehl[2], Hugues Matile[3], Hongli Hu[1,2], Daniel Hilger[1], Gebhard F.X. Schertler[4], Aashish Manglik[5,6], Georgios Skiniotis [1,2], Roger J.P. Dawson [3] & Brian K. Kobilka[1]

Single-particle cryo-electron microscopy (cryo-EM) has recently enabled high-resolution structure determination of numerous biological macromolecular complexes. Despite this progress, the application of high-resolution cryo-EM to G protein coupled receptors (GPCRs) in complex with heterotrimeric G proteins remains challenging, owning to both the relative small size and the limited stability of these assemblies. Here we describe the development of antibody fragments that bind and stabilize GPCR-G protein complexes for the application of high-resolution cryo-EM. One antibody in particular, mAb16, stabilizes GPCR/G-protein complexes by recognizing an interface between Gα and Gβγ subunits in the heterotrimer, and confers resistance to GTPγS-triggered dissociation. The unique recognition mode of this antibody makes it possible to transfer its binding and stabilizing effect to other G-protein subtypes through minimal protein engineering. This antibody fragment is thus a broadly applicable tool for structural studies of GPCR/G-protein complexes.

[1] Department of Molecular and Cellular Physiology, Stanford University School of Medicine, 279 Campus Drive, Stanford, CA 94305, USA. [2] Department of Structural Biology, Stanford University School of Medicine, 279 Campus Drive, Stanford, CA 94305, USA. [3] Roche Pharma Research and Early Development, Therapeutic Modalities, Roche Innovation Center Basel, F.Hoffmann-La Roche Ltd, Grenzacherstrasse 124, 4070 Basel, Switzerland. [4] Laboratory of Biomolecular Research, Paul Scherrer Institute, 5232 Villigen, Switzerland. [5] Department of Pharmaceutical Chemistry, University of California San Francisco, 1700 4th Street, San Francisco, CA 94143, USA. [6] Department of Anesthesia and Perioperative Care, University of California San Francisco, 1700 4th Street, San Francisco, CA 94143, USA. Correspondence and requests for materials should be addressed to R.J.P.D. (email: roger.dawson@roche.com) or to B.K.K. (email: kobilka@stanford.edu)

G-protein coupled receptors (GPCRs) make up the largest receptor family in the human genome, comprising around 800 members. GPCRs are expressed ubiquitously and play essential roles of signal transduction in response to a wide variety of extracellular stimuli such as photons, ions, neurotransmitters, hormones and proteins. Given their numerous physiological roles, GPCRs are implicated in numerous diseases and ~30% of marketed drugs are targeting this receptor family[1]. Recent advances in GPCR crystallography have led to high-resolution structures of G-protein[2] and arrestin[3] complexes, which have enhanced our understanding of the structural details underlying ligand binding and signal transduction at the atomic level. The first crystal structure of a GPCR/G-protein complex was that of the $\beta_2$ adrenergic receptor in complex with stimulatory G-protein, $G_s$ ($\beta_2AR/G_s$)[2]. This was later followed by the crystal structure of $A_{2A}$ adenosine receptor in complex with miniG$_s$ ($A_{2A}R/miniG_s$) in which a highly engineered G$\alpha_s$ that consists of only the G$\alpha$ ras-like domain was used in place of the full heterotrimer[2,4]. The fact that such drastic protein engineering is needed to obtain diffraction quality crystals reflects the difficulty inherent in GPCR–G-protein complex crystallography. Despite the technological advancement, crystallographic studies of these complexes remains extremely difficult. More recently, single-particle cryo-electron microscopy (cryo-EM) has emerged as an alternative technique with the ability to provide near-atomic resolution maps, as demonstrated for two class B GPCRs[3,5,6] both in complex with $G_s$: the glucagon-like peptide1 receptor/$G_s$ (Glp-1R/$G_s$)[6,7] as well as the calcitonin receptor/$G_s$ (CTR/$G_s$)[5]. These studies have highlighted the possibility of employing cryo-EM to obtain the structures of GPCR-G protein complexes. Compared to class A GPCRs, class B receptors include a structured extracellular domain that may aid in particle alignment. Furthermore, for Gs proteins, Nb35[2] stabilizes these complexes against GTP$\gamma$S by stabilizing an interface between the Ras-like domain of the G$\alpha_s$ subunit and the G$\beta$ subunit. These factors make class B GPCR/Gs protein complexes more tractable targets for cryo-EM compared to class A or other G-protein subtype complexes.

Apart from the GPCR/$G_s$ complex, the only structure available at high-resolution has been limited to the MetaII rhodopsin/G$\alpha_{Ct}$ where the last 11-amino-acid fragment of $G_{transducin}$ was co-crystalized with the activated rhodopsin[8]. Although in silico analyses using this complex have provided insights into the conformational changes that allow $G_i$ coupling as well as general principles for G protein coupling specificity[9,10], experimental structures of other G-protein complexes are invaluable to understand how receptors selectively engage one G-protein subtype over others. G-protein mimetic nanobodies have been used as a surrogate to capture the active conformation of a receptor[11–14], but it may require an extensive effort to find such nanobodies and the trapped conformation may not necessarily represent the G-protein engaged state.

Here we describe the development of an antibody, termed mAb16, that recognizes the heterotrimeric $G_{i/o}$ type G protein and enhances the stability of GPCR-Gi/o complexes, while simultaneously adding an asymmetric feature that may aid with cryo-EM particle projection alignment. As antibodies typically bind to their targets in a rigid manner, such an antibody would be expected to enable structure determination of GPCR/G-protein complexes by cryo-EM. mAb16 recognizes a unique epitope, binding at the interface between the $\alpha$ and $\beta$ subunits of heterotrimeric $G_i$. While the antibody confers extra stability to GPCR/$G_{i/o}$ complex as well as increased resistance to GTP$\gamma$S-triggered dissociation of the complex in a manner similar to Nb35 for $G_s$, mAb16 and Nb35 bind to completely different epitopes. We have recently succeeded in obtaining a near-atomic resolution map of the mu-opioid receptor ($\mu$OR)/$G_i$ complex using this antibody fragment[15]. Although this antibody is specific against $G_{i/o}$-family G-proteins, its ability to bind and stabilize the heterotrimer can be transferred to other G-protein subtypes through a simple protein engineering strategy.

## Results

**Selection of monoclonal antibodies.** Despite exhaustive attempts to crystalize a complex between rhodopsin and heterotrimeric $G_{i1}$[16], we were unsuccessful in producing diffraction quality crystals. We presumed that this was due to the flexibility of the alpha-helical domain of G$\alpha_{i1}$ as this domain separates from Ras-like domain and becomes flexible upon receptor-mediated activation in the nucleotide-free state[17,18]. We then set out to discover antibodies that could reduce this flexibility and facilitate crystallographic and cryo-EM structural studies of the complex. Mice were immunized with purified rhodopsin/$G_{i1}$ complex and hybridoma cells were prepared from the isolated mice splenocytes. Clones that showed enzyme-linked immunosorbent assay (ELISA) and immunoprecipitation positive reaction were screened further using an analytical size-exclusion chromatography (SEC) assay with purified monoclonal antibodies. Most of the SEC-positive clones were G$\beta\gamma$-binders, reflecting that it is the most stable and rigid component of the complex (Fig. 1a, c). Interestingly, we found a single clone that binds and confers GTP$\gamma$S resistance to the rhodopsin/$G_{i1}$ complex (Fig. 1a, b). Based on the clone identification number, we named its antibody mAb16. Notably, this clone does not show binding to any single component of $G_{i1}$, but binds specifically to the intact heterotrimeric form of $G_{i1}$, suggesting that it binds a composite epitope at the interface between G$\alpha_{i1}$ and G$\beta\gamma$ subunits (Fig. 1c). We provide amino-acid sequence of mAb16 in the Supplementary Note 1.

**Crystal structure of Gi1/scFv16.** In order to better understand the recognition mode of mAb16, we crystallized a fully soluble heterotrimeric $G_{i1}$ in complex with mAb16 fragments. We tried both a Fab fragment (Fab16) and single-chain variable fragment (scFv16) derived from mAb16. Both Gi1/Fab16 and Gi1/scFv16 complexes formed crystals but only the scFv16 version diffracted to high resolution, presumably due to the intrinsic flexibility of the linker between the variable and the constant domain of the Fab[19]. The crystal structure of the $G_{i1}$/scFv16 complex was solved at 2.0 Å by molecular replacement using $G_{i1}$ (PDB ID: 1GP2) and an scFv fragment (PDB ID: 4NKD) as search models (Table 1). The overall structure of $G_{i1}$ in complex with scFv16 is very similar to $G_{i1}$ alone (Fig. 2a). The relative position of Ras-like domain and alpha-helical domain of G$\alpha_{i1}$ moves closer to G$\beta_1$ by a small rotation movement around the $\alpha$N-b1 junction (Fig. 2b). This slight movement leads to two additional interactions between Thr182 of G$\alpha$i1 and Asn119 of G$\beta_1$, and Arg205 of G$\alpha$i1 and Thr143 of G$\beta_1$, located in Switch I and Switch II region, respectively (Fig. 2b). This could be due to the tighter association between these two subunits mediated by scFv16, although it may be the consequence of different crystal contacts between $G_{i1}$ alone and $G_{i1}$/scFv16. The structure of the $G_{i1}$/scFv16 complex shows that scFv16 recognizes an epitope composed of the terminal part of the $\alpha$N helix of G$\alpha_{i1}$ as well as part of the G$\beta_1$ subunit (Fig. 2c, Supplementary Fig. 1). The complementarity-determining region 3 of the heavy chain (CDR-H3) extends to interact with G$\beta_1$ with its tip and G$\alpha_{i1}$ with its side. CDR2-H2 and CDR-H1 support the interaction with G$\alpha_{i1}$ and G$\beta_1$, respectively by making hydrogen bonds and van der Waals contacts. CDR-L1 is exceptionally long and makes extensive contact with the edge of the $\alpha$N helix together with CDR-L3 (Fig. 2c). There is no obvious interaction between scFv16 and G$\gamma_2$ subunit.

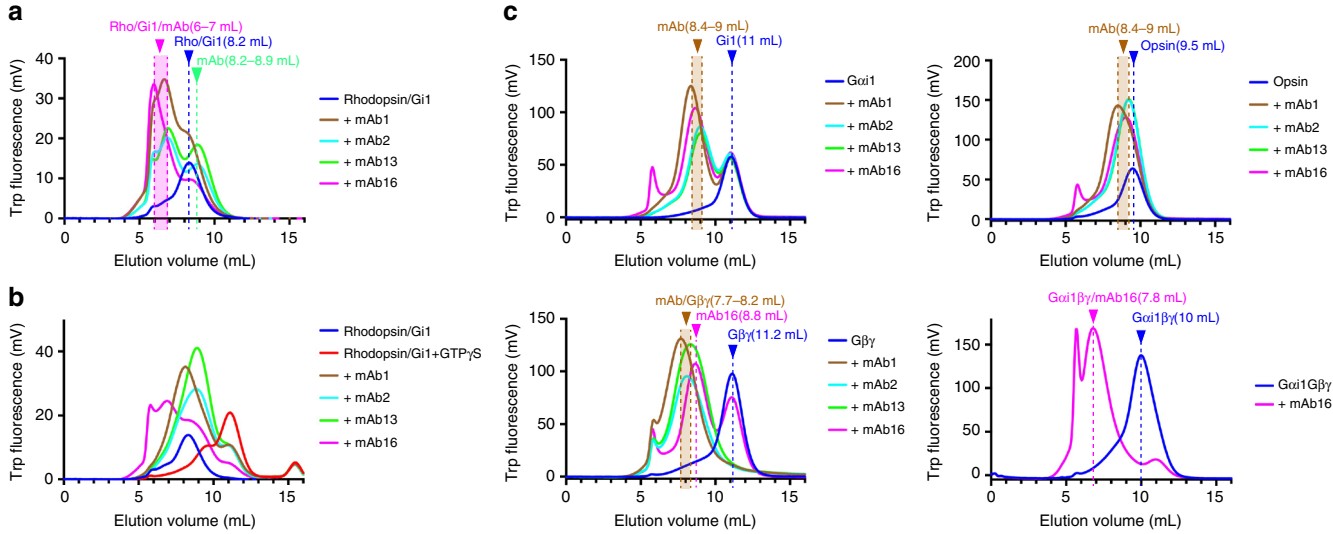

**Fig. 1** Isolation of mAb16 and its binding profile to each component. **a** Analytical SEC of rhodopsin/$G_{i1}$ with each antibody. Rhodopsin/$G_{i1}$ runs at 8.2 mL and each mAb alone runs 8.4–9 mL (**c**). Rhodopsin/$G_{i1}$ bound to mAb makes higher molecular weight product and migrates at the elution volume of 6–7 mL depending on the mAb. **b** Analytical SEC of rhodopsin/$G_{i1}$ with each antibody following to GTPγS treatment. Intact complex remains at 6–7 mL only in the mAb16 condition. **c** Analytical SEC of individual component of rhodopsin/$G_{i1}$ or heterotrimeric $G_{i1}$ with each antibody. Top left: Binding experiment with $G\alpha i1$ subunit and each mAb. The peak of $G\alpha i1$ at 11 mL stays intact indicating there is no binding with each mAb. Top right: Binding experiment with opsin. Both mAb peaks and Opsin peak (at 9.5 mL) stays intact. Bottom left: Binding experiment with $G\beta\gamma$ subunit. The peak of $G\beta\gamma$ at 11.2 mL disappears upon incubating with mAbs except mAb16 and each mAb peak shifts towards left compared to the ones with $G\alpha i1$ or opsin indicating those mAbs recognize $G\beta\gamma$ subunit as an epitope. Bottom right: Binding experiment with heterotrimeric Gi1

### Table 1 Data collection and refinement statistics

|  | Gi1/scFv16[a] |
| --- | --- |
| *Data collection* | |
| Space group | $P222_1$ |
| Cell dimensions (σ) | |
| $a$, $b$, $c$ (Å) | 58.51, 104.74, 211.82 |
| $\alpha$, $\beta$, $\gamma$ (°) | 90.00, 90.00, 90.00 |
| Resolution (Å) | 39.26–2.00(2.07–2.00)[b] |
| $R_{sym}$ or $R_{merge}$ | 0.176(0.888) |
| $I/\sigma I$ | 9.91(0.73) |
| Completeness (%) | 99.16(99.47) |
| Redundancy | 4.6(4.8) |
| *Refinement* | |
| Resolution (Å) | 39.26–2.00(2.07–2.00) |
| No. reflections | 88,191(8710) |
| $R_{work}/R_{free}$ | 0.1746/0.2097(0.2682/0.2940) |
| No. atoms | |
| Protein | 7567 |
| Ligand/ion | 41 |
| Water | 628 |
| *B-factors* (Å$^2$) | |
| Protein | 47.08 |
| Ligand/ion | 31.55 |
| Water | 51.44 |
| R.m.s. deviations | |
| Bond lengths (Å) | 0.007 |
| Bond angles (°) | 1.19 |

[a]The data set was collected from one single crystal
[b]Values in parentheses are for highest-resolution shell

As the α subunits of all $G_{i/o}$ family members have high sequence similarity at the epitope residues in the αN helix (Fig. 3a) and can form a complex with $G\beta_1\gamma_2$, we expected that Fab16 would bind to all $G_{i/o}$ family proteins. Using analytical fluorescent SEC, we show that Fab16 can bind to five different $G_{i/o}$ type G-proteins but not $G_s$, as it has poor sequence similarity to $G_{i/o}$ members at this epitope region (Fig. 3b).

**Application to other GPCR-$G_{i/o}$ complexes.** Since Fab16 was initially isolated as a stabilizing agent that confers resistance to GTPγS triggered dissociation to a rhodopsin/$G_{i1}$ complex and was later found to bind to a panel of $G_{i/o}$ family members, we investigated whether it confers the same GTPγS resistance to other $G_{i/o}$ type GPCR complexes. We chose a μ-opioid receptor/$G_{i1}$ (μOR/$G_{i1}$) and an $M_2$ muscarinic acetylcholine receptor/$G_{oA}$ ($M_2R/G_{oA}$) as representative family A $G_{i/o}$-coupling GPCR complexes. Purified μOR/$G_{i1}$ or $M_2R/G_{oA}$ complex solubilized in detergent was incubated with GTPγS in the presence or absence of Fab16, then analysed for dissociation by analytical SEC. Both μOR/$G_{i1}$ and $M_2R/G_{oA}$ complexes showed a leftward peak shift upon incubating with Fab16, indicating its binding and also became GTPγS resistant, as they showed much less dissociation in the presence of Fab16 than the complex alone (Fig. 3c). These data with GPCR/$G_{i/o}$ complexes are consistent with the binding experiments of Fab16 with G-protein alone, and indicate that it stabilizes $G_{i/o}$ type GPCR complexes in general. In order to show the applicability of scFv16 to structural analysis of GPCR-G protein complexes, we have recently solved a near-atomic resolution map of μOR/$G_{i1}$ complex using scFv16[15]. The presence of scFv16 enhanced complex stability towards specimen vitrification for cryo-EM, thereby enabling quality single-particle reconstructions.

**Influence of Fab16 on nucleotide binding.** In order to further investigate mAb16 for its protection mechanism against GTPγS, we monitored the binding kinetics of GTPγS to nucleotide-free $M_2R/G_{oA}$ complex. In the absence of Fab16, BODIPY-FL-GTPγS, a fluorescent analogue of GTPγS, binds to the complex with fast kinetics reflecting its ability to bind and trigger the dissociation of the complex. In contrast, BODIPY-FL-GTPγS binds to $M_2R/G_{oA}/$ Fab16 complex ~70 times slower and to a much lower extent

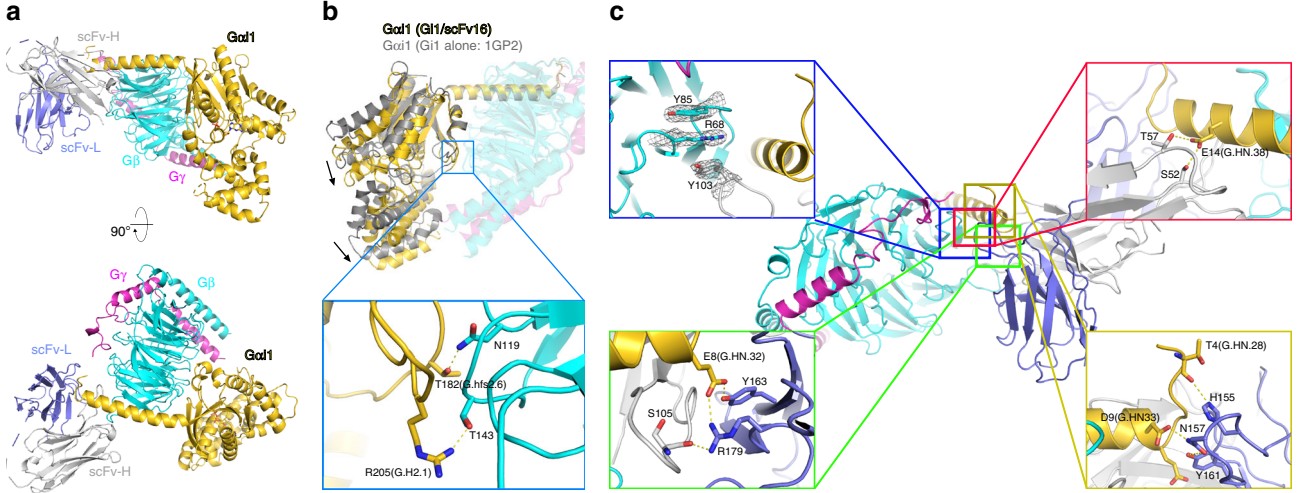

**Fig. 2** Crystal structure of G$_{i1}$/scFv16 and characterization of Fab16. **a** Overall structure of Gi1/scFv16 complex. Cartoon representation with Gα$_{i1}$ in gold, Gβ in cyan, Gγ in magenta, scFv-heavy chain in light grey and scFv-light chain in light blue. **b** Superposition of G$_{i1}$/scFv16 structure onto G$_{i1}$ (PDB: 1GP2) based on alignment of Gβγ subunits. Gα$_{i1}$ (1GP2) in grey and G$_{i1}$/scFv16 in the same colour code as in **a**. For clarity, Gβγ subunits and scFv16 is shown as transparent cartoon. Arrows show a slight rotational displacement of Gα$_{i1}$ towards Gβ$_1$ compared to Gi1 alone. Additional interactions are formed between switch I and switch II of Gα$_{i1}$ and Gβ$_1$. **c** Interaction between G$_{i1}$ and scFv16. The residues participating in the interactions are depicted with stick models in the expanded panels. Residue numbers are shown with Common Gα Numbering (CGN) code for Gα$_{i1}$[42]

(Fig. 4a). On the other hand, the binding of GDP is only modestly affected when the complex is bound to Fab16 (Fig. 4a). Next we examined the basal nucleotide exchange rate of Gi1 and Gi1/ Fab16 under conditions where GDP release is rate-limiting (Fig. 4b). Fab16-bound Gi1 releases GDP ~1.5 to 2-fold slower than Gi1 alone (Table 2). These findings suggest that in the absence of a coupled GPCR, Fab16 stabilizes the heterotrimer in its GDP-bound conformation. As a comparison, we examined the effect of Nb35 that was originally developed against the β$_2$AR/G$_s$ complex[2]. Structurally, it binds at the interface between switch II and helix III of Gαs and Gβ$_1$, and has been characterized to provide GTPγS resistance to a GPCR/G$_s$ complex. It's been used for structural studies of all GPCR/G$_s$-type complexes both in crystallography and cryo-EM so far[2,5–7]. We monitored the binding kinetics of GTPγS and GDP to the nucleotide-free β$_2$AR/ G$_s$ complex. BODIPY-FL-GTPγS binds to the complex with fast kinetics in the absence of Nb35 but becomes extremely slow or negligible when bound with Nb35 (Fig. 4c), presumably due to the inhibition of the conformational change in switch II and switch III. BODIPY-FL-GDP showed no detectable binding to the complex alone or to the complex bound with Nb35 (Fig. 4c).

**Generalization of mAb16 binding to other G-protein subtypes.** Because mAb16's epitope is located on a short stretch of the αN helix of the Gα subunit (Figs. 2a, b and 3a) as well as a small part of the Gβ$_1$ subunit that can complex with all Gα-protein subtypes, we sought to engineer the αN helix of the other G-protein α-subunits in order to generalize scFv16 binding to all G-protein subtypes. Starting with the Gα$_s$ subunit, we generated a chimera in which the αN helix was replaced by the equivalent region of Gα$_i$ (Gα$_{si}$N; residues 1–38 of Gαs replaced by residues 1–31 of Gαi1) (Fig. 5a). The Gα$_{si}$N protein forms a heterotrimer with the Gβγ subunit and forms a stable complex with the β$_2$AR (Fig. 5b). The β$_2$AR/G$_{si}$N complex can bind Fab16 and is largely protected from dissociation induced by GTPγS, whereas β2AR/GsiN complex alone dissociated almost completely under the same conditions (Fig. 5b). We then used the same engineering approach to transfer Fab16 binding ability to G$_{11}$, a G$_q$ family member, and replaced the αN helix of Gα$_{11}$ with that from Gα$_{i1}$ (Gα$_{11i}$N, residues 1–35 of Gα$_{11}$ replaced by residues 1–29 of Gα$_{i1}$) (Fig. 5a).

The Gα$_{11i}$N protein forms a heterotrimer with Gβγ subunit and couples to form a stable complex with the M$_1$ muscarinic acetylcholine receptor (M$_1$R) (Fig. 5c). The M$_1$R/G$_{11i}$N complex alone dissociates upon incubation with GTPγS, whereas the M$_1$R/ G$_{11i}$N complex bound to Fab16 showed GTPγS resistance (Fig. 5c) consistent with G$_{i/o}$ and G$_{si}$N complexes. Negative stain EM visualization of the M$_1$R/G$_{11i}$N complex reveals a monodisperse sample (Fig. 5d). These results demonstrate that Fab16 (or scFv16) can be used as a tool with broad variety of GPCR/G-protein complexes by substituting the αN helix of other Gα-subunit with equivalent region of Gαi1. Scanning the chimera junction between Gα$_{11}$ and Gα$_{i1}$ shows that a smaller substitution in the middle of the αN helix is still tolerated for the expression and the heterotrimer formation (Supplementary Fig. 2a). When replaced with the equivalent residues with this minimal chimeric region (residues 1–18 of Gαi1), both G$_s$ and G$_{11}$ are enabled to bind Fab16 and form stable complexes with respective GPCRs (Supplementary Fig. 2b, c.) The same minimal region when transferred to G$_{12}$ also enables Fab16 binding (Supplementary Fig. 2d). The binding interface of Gβ1 subunit to scFv16 is limited compared to Gα subunit in the crystal structure. These residues are mostly conserved among Gβ family members except Gβ5 (Supplementary Fig. 3). There is no direct interaction between Gγ2 and scFv16 in the crystal structure. The fact that mAb16 was originally raised against and indeed binds to Rhodopsin/Gi1 that is composed of Gγ1 from the native bovine retina and still binds G-proteins or GPCR/G-protein complexes composed of Gγ2 indicates that Fab/scFv16 binds to the heterotrimeric G-protein regardless of the composition of the γ-subunit. Therefore, the binding ability to Fab16 is transferable to broad range of G-protein family members with minimal chimeric constructs. We provide amino-acid sequences of G-protein chimera constructs in the Supplementary Note 2 as well as the primers used for the construction in the Supplementary Table.

**Discussion**

In this work, we have developed a unique antibody fragment that recognizes an interface on heterotrimeric G$_{i1}$. The antibody confers the GPCR–G$_{i/o}$ complexes the resistance to GTPγS-induced dissociation. This property is also observed with formerly

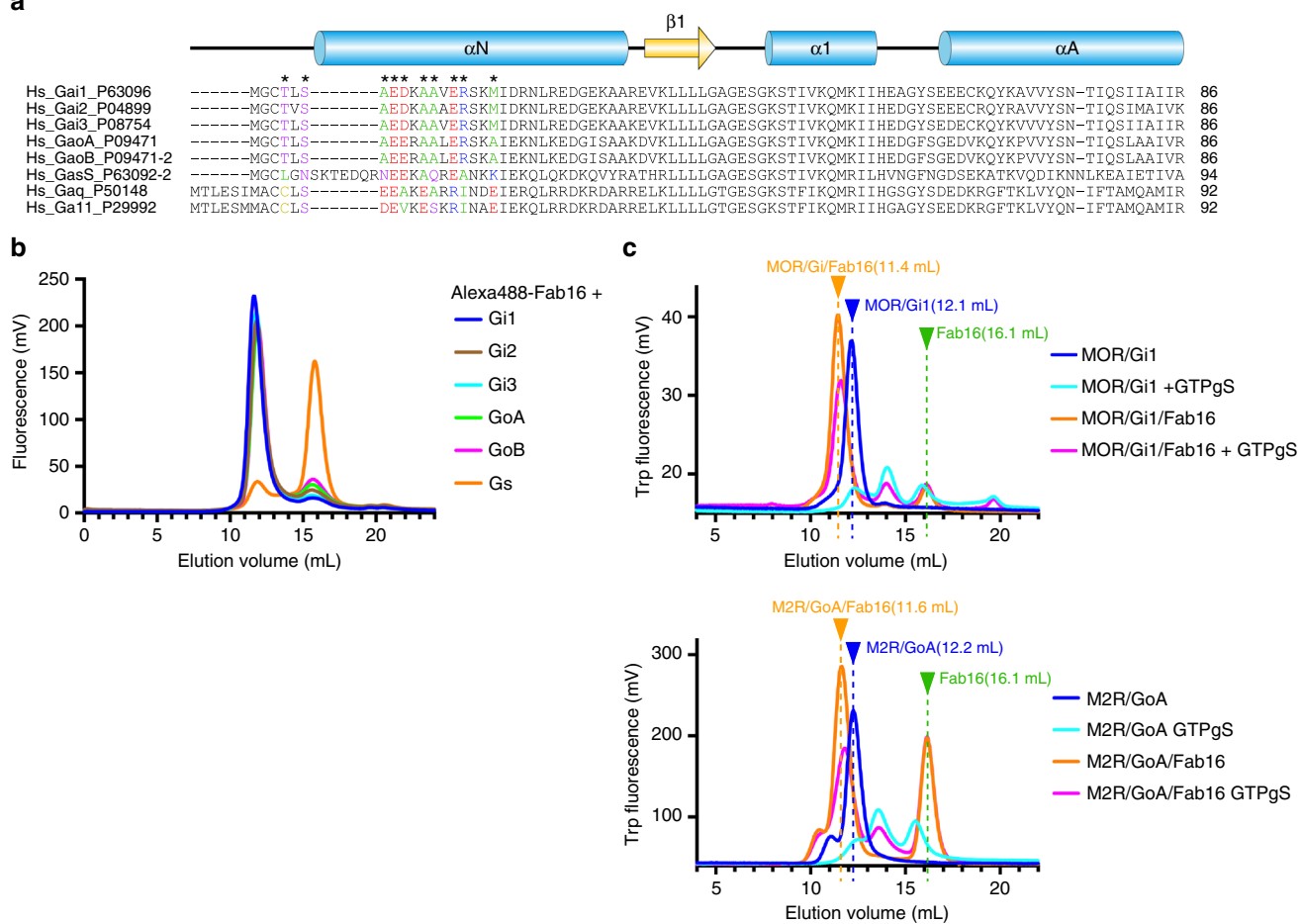

**Fig. 3** Sequence alignment of G-protein family members and binding profile of Fab16. **a** Multiple sequence alignment of amino-termini of representative Gα subunits from human. UniProt numbers are provided after each G-protein subtype name. Secondary structures are shown as cylinder (helix) and arrow (strand). The asterisks indicate the residues in contact with scFv16 in Gα$_{i1}$ and those corresponding residues are coloured according to their property: Positive in blue, negative in red, hydrophobic in green, polar in purple, cysteine in yellow. **b** Fluorescent SEC analysis of binding of the fluorescently labelled Fab16 with G-protein family members. **c** Analytical tryptophane fluorescent SEC of μOR/G$_{i1}$ and M$_2$R/G$_{oA}$ with GTPγS in the presence or absence of Fab16. Each complex alone runs around 12.2 mL. Upon binding to Fab16, they run at 11.4 mL or 11.6 mL indicating the binding of Fab16 to these GPCR/G-protein complexes. Excess free Fab16 runs at 16.1 mL. Dissociated components upon incubating with GTPγS show smaller peaks at 13.5–16 mL

identified Nb35 for Gs complexes[2]. Nb35 and mAb16 engage distinct epitopes at the G-protein interface: Nb35 binds at the switch II and αIII helix of Gα$_s$ and Gβ, while mAb16 engages the αN helix of Gα$_{i1}$ and Gβ. The switch II region adopts a distinct conformation upon binding of GTPγS compared to the nucleotide-free or GDP-bound state observed in the crystal structures as well as in the EPR spectroscopic measurement[2,20–23]. Nb35 is reported to suppress nucleotide exchange turnover of the CTR/G$_s$ complex[5] and indeed it prevents GTPγS binding to the β2AR/G$_s$ complex (Fig. 4c), which we presume due to the fixed conformation of the switch II in the nucleotide-free state and steric clash of the switch III with Nb35. On the other hand, mAb16 binds 40 Å away from the nucleotide-binding pocket with no direct contact with this region yet prevents the binding of GTPγS to the nucleotide-free GPCR/G-protein complex (Fig. 4a) and helps heterotrimeric G$_{i1}$ to retain GDP in the nucleotide-binding pocket (Fig. 4b). It has been reported that heterotrimer formation rigidifies the switch II region that is in direct contact with Gβ subunit[23]. Thus, by stabilizing interactions between the Gβ subunit and switch II, mAb16 prevents GTPγS binding to the empty pocket. This may also explain the slower GDP release as Gβ subunit functions as a GDP dissociation inhibitor (GDI)[24]. Additional interactions formed between switch I, II and Gβ$_1$

subunit upon scFv16 binding supports this idea (Fig. 2b). Another possibility would be inferred from HDX-MS measurements that revealed the dynamic nature of αN-β1 junction of G$_s$ during the complex formation with β2AR and the dissociation upon addition of GDP/AlF4[25]. Binding of mAb16 would tighten the association of αN helix with Gβ subunit and concomitantly reduce the dynamics of αN–β1 junction, which may influence the binding of nucleotide through the β1-strand and the P-loop that is in direct contact with nucleotide. Previous work showing that binding of GTPγS triggers release of αN helix from Gβ and promotes its unfolding[26] indicates that there could be an allosteric effect between the nucleotide-binding pocket and αN helix.

Contrary to the large diversity of different GPCR genes in the human genome, there are only four major G-protein family members: G$_s$, G$_{i/o}$, G$_{q/11}$ and G$_{12/13}$[27]. Among them G$_{i/o}$ is the most broadly coupling G-protein[10]. mAb16 is originally raised against rhodopsin/G$_{i1}$ complex and binds rhodopsin/G$_{i1}$, μOR/G$_{i1}$ and M$_2$R/G$_{oA}$ complexes, suggesting that it likely binds all G$_{i/o}$ type complexes, according to the sequence similarity of the epitope residues and the binding profile of individual G-protein alone (Fig. 3b). In contrast, the binding interface of Gα$_s$ to Nb35 is not conserved among Gα subtypes; therefore it would require elaborate protein engineering to transfer the binding surface to

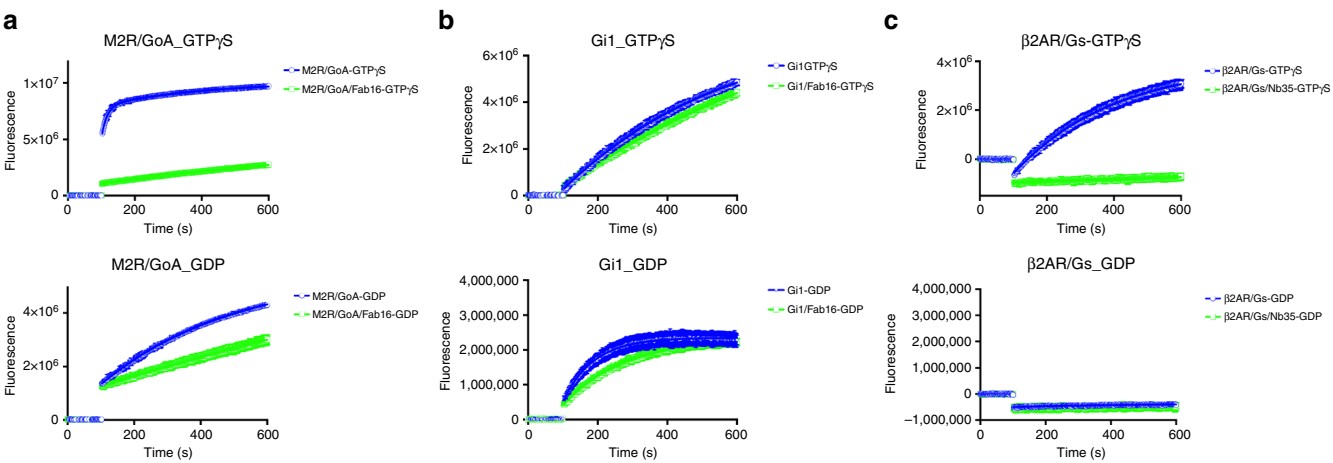

**Fig. 4** Nucleotide-binding kinetics. **a** Influence of Fab16 on the nucleotide-binding kinetics of the purified $M_2R/G_{oA}$ complex. Nucleotide binding was monitored by using BODIPY-FL-GTPγS or BODIPY-FL-GDP. **b** Influence of Fab16 on the nucleotide release kinetics of $G_{i1}$. GDP release was monitored by BODIPY-FL-GTPγS- or BODIPY-FL-GDP-binding kinetics under conditions where GDP release is the rate-limiting step. **c** Nucleotide binding to β2AR/$G_s$ complex in the presence or absence of Nb35. The curves represent the mean ± standard error of three experiments

## Table 2 Effect of Fab16 or Nb35 on the nucleotide-binding/releasing rate

|  | **M2R/GoA + GTPγS** | **M2R/GoA + Fab16 + GTPγS** | **M2R/GoA + GDP** | **M2R/GoA + Fab16 + GDP** |
|---|---|---|---|---|
| $k$ (1/s) | 0.061 ± 0.002 (fast)[a] | 0.00092 ± 0.00004 | 0.00244 ± 0.00003 | 0.00042 ± 0.00009 |
| $k$ (1/s) | 0.0032 ± 0.0002 (slow)[a] | | | |
|  | **Gi1 + GTPγS** | **Gi1 + Fab16 + GTPγS** | **Gi1 + GDP** | **Gi1 + Gab16 + GDP** |
| $k$ (1/s) | 0.00206 ± 0.00003 | 0.00116 ± 0.00003 | 0.0115 ± 0.0002 | 0.00572 ± 0.00007 |
|  | **β₂AR/G$_s$ + GTPγS** | **β₂AR/G$_s$ + Nb35 + GTPγS** | **β₂AR/G$_s$ + GDP** | **β₂AR/G$_s$ + Nb35 + GDP** |
| $k$ (1/s) | 0.00327 ± 0.00005 | ND | ND | ND |

All values are expressed as mean ± s.e.m. of triplicate experiments
[a]The binding kinetics was best fit and analysed by two phase association. The fraction corresponding to the fast component is 0.526 ± 0.005

other family members or to evolve the nanobody itself. On the other hand, the binding surface of $G\alpha_{i1}$ to mAb16 is located in a small stretch of the αN helix. Since the αN helix apparently serves as a separate module from the Ras-like domain to interact with Gβγ, it is more amenable to generating functional chimeras[28–30]. Previous studies have shown that the entire αN helix of $G\alpha_q$, $G\alpha_{12}$ and $G\alpha_{13}$ can be substituted with the corresponding region of $G\alpha_{i1}$ to produce functional chimeric $G_{i/q}$, $G_{i/12}$ and $G_{i/13}$. These engineered G proteins retained the biochemical properties of their counterpart wild types, while αN helix of $G\alpha_{12/13}$ is reportedly important for the receptor selectivity[31]. Chimeric Gs protein has also been made with various lengths of αN from Gαi subunit to investigate the functional role of this region[32,33]. While the chimeras exhibit a large constitutive activity when the substitution goes beyond αN helix to replace the residues 1–62 of $G\alpha_s$ by 1–54 of $G\alpha_i$, a minor increase in the basal activity was observed when residues 1–41 of $G\alpha_s$ is replaced by 1–34 of $G\alpha_i$. Our substitutions of $G_{siN}$ and $G_{11iN}$ are both within the range of these chimeric designs and therefore would be expected to behave in the same way as the wild-type counterparts. In fact, our G protein chimeras form functional heterotrimers with co-expressed Gβγ subunit and form stable and functional complexes with cognate GPCRs. The more conserved chimeras where the residues 1–18 of $G\alpha_{i1}$ is transferred to the equivalent residues of $G\alpha_s$, $G\alpha_{11}$ and $G\alpha_{12}$ are also able to bind with Fab16 (Supplementary Fig. 2).

In conclusion, the antibody fragment derived from mAb16 promotes the stabilization of GPCR/G-protein complexes and adds an asymmetric feature that may aid with cryo-EM particle projection alignment. The usefulness of the antibody fragment in structural determination was proven by the cryo-EM structure of

μOR/$G_{i1}$ complex where the presence of scFv16 stabilized the complex for high-resolution cryo-EM work. Furthermore, this antibody fragment can be applied to other G-protein subtypes with minimal protein engineering and therefore would be expected to be a broadly applicable tool for cryo-EM studies of any GPCR/G-protein complex.

## Methods

**Protein expression and purification**. Rhodopsin/$G_{i1}$ complex was purified as described previously[16]. Briefly, bovine rhodopsin with three mutations, N2C, M257Y and N282C, was stably expressed in and purified from HEK293S GnTI⁻ cells using 1D4 immunoaffinity chromatography. Purified rhodopsin was incubated with $G_{i1}$ reconstituted from recombinant Gαi1 subunit from *Escherichia coli* BL21 (DE3) cells (Novagen) and Gβγ subunit purified from bovine retina (W L Lawson Company). Rhodopsin/$G_{i1}$ complex formation was triggered by the irradiation through 495 nm long-pass filter in the presence of apyrase (Sigma-Aldrich). Rhodopsin/$G_{i1}$ complex was separated from the free rhodopsin or $G_{i1}$ by SEC on a Tricorn 10/600 column packed with Superdex 200 (GE healthcare) in a buffer containing 100 mM NaCl, 20 mM Hepes pH 7.5, 0.01% lauryl maltose neopentyl glycol (MNG), 2 mM 2-mercaptoethanol.

μOR with a cleavable amino and carboxy-terminal FLAG- and His-tag[13] was expressed in *Spodoptera frugiperda* Sf9 insect cells using baculovirus infection system (Expression Systems). Cells were solubilized in 1% n-dodecyl-β-D-maltoside (DDM) (Anatrace), 0.2% 5-cholesterol hemisuccinate (CHS) (Steraloids) and the soluble fraction was purified by Ni-chelating sepharose chromatography. The eluted protein was supplemented with 2 mM CaCl2, loaded onto M1 anti-FLAG immunoaffinity column (prepared in house) and washed with progressively lower concentrations of the antagonist naloxone (Sigma-Aldrich). Receptor was eluted in a buffer consisting of 100 mM NaCl, 20 mM Hepes pH 7.5, 0.1% DDM, 0.01% CHS with 50 nM naloxone, and further purified by SEC on a Superdex 200 10/300 column in a buffer containing 1 μM lofentanil (Tronto Research Chemicals) to exchange the ligand. Monomeric fractions were pooled, further supplemented with a twofold molar excess of lofentanil and concentrated to ~100 μM for complex formation.

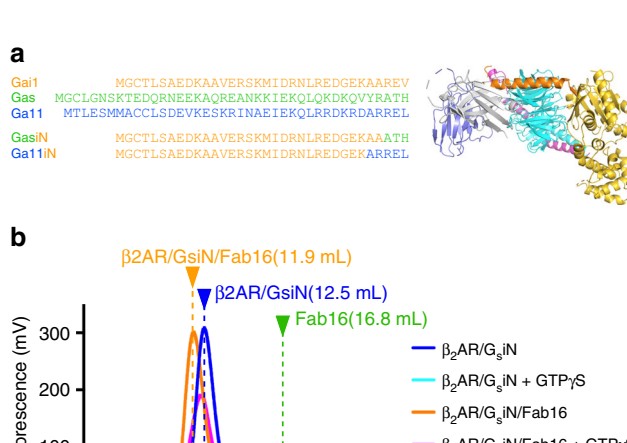

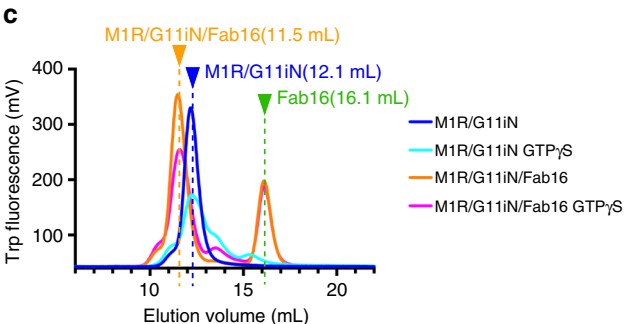

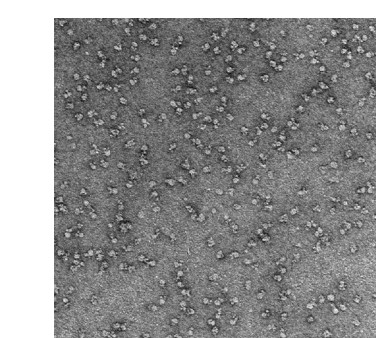

**Fig. 5** Generation of chimeric G-proteins. **a** Alignment of the αN helix of the G-protein subfamilies and the sequence of the chimeric Gα subunits. Transferred region from Gα$_{i1}$ in each chimera is colored in orange. **b**, **c** Analytical SEC of β2AR/G$_{siN}$ and M$_1$R/G$_{11iN}$ complexes incubated with GTPγS in the presence or absence of Fab16. Protein elution profiles were monitored by the intrinsic tryptophan fluorescence. **d** Negative stain electron microscopy image of purified the M$_1$R/G$_{11iN}$/scFv16 complex

β2AR was purified in the same way as described previously[34]. Briefly, Sf9 insect cells were lysed by osmotic shock prior to solubilization of the membrane fraction by DDM. Solubilized receptor was first purified by M1 anti-FLAG immunoaffinity chromatography, followed by alprenolol-sepharose chromatograpy (alprenolol-sepharose resin prepared in house) to isolate only functional receptor. Alprenolol-sepharose eluate was concentrated on M1 FLAG affinity resin, and then washed with ligand-free buffer for 1 h at room temperature to eliminate bound alprenolol. Receptor was eluted in a buffer consisting of 20 mM Hepes pH 7.5, 350 mM NaCl, 0.1 % DDM and 0.01% CHS and further purified by size-exclusion chromatography on a Superdex 200 10/300 column (GE Healthcare) in buffer containing 20 mM Hepes pH 7.5, 100 mM NaCl, 0.05 % DDM, 0.005% CHS and 1 µM BI-167107 (Boehringer-Ingelheim). The eluted receptor was concentrated to ~100 µM for complexing.

For M$_1$R and M$_2$R, we used constructs used in previous studies[12,35] with some modifications. Briefly, T4L in the ICL3 of M$_1$R was removed, and residues 219–232

and 345–354 were added to TM5 and TM6, respectively. The ICL3 of M$_2$R was extended by 15 amino-acid residues from TM6. Primer sequences for these modifications are provided in the Supplementary Table. The amino-acid sequences for these receptors are provided in the Supplementary Note 2. These receptors were purified essentially in the same way as µOR with detergent exchange to MNG during M1 FLAG chromatography and using atropine (Sigma-Aldrich) and iperoxo (Sigma-Aldrich) in place of naloxone and lofentanil, respectively.

Heterotrimeric G-proteins were expressed and purified as previously described. Briefly, *Trichuplusia ni* (Hi5) insect cells (Expression Systems) were co-infected with two viruses, one encoding the wild-type human Gα subunit and another encoding the wild-type human β1γ2 subunits with an decahistidine tag inserted at the amino terminus of the β1 subunit with HRV-3C protease cleavable site. In the case of G$_{11iN}$, additional virus encoding Ric8A was also co-infected. Cells were harvested 48 h post infection, lysed in hypotonic buffer and lipid-modified heterotrimeric G-protein was extracted in a buffer containing 1% sodium cholate (Sigma-Aldrich) and 0.05% DDM. The soluble fraction was purified using Ni-chelating sepharose chromatography, and the detergent was exchanged from cholate/DDM mixture to DDM alone. After elution, HRV-3C protease (in-house prepared) was added and the protein was dialyzed against a buffer containing 20 mM Hepes pH 7.5, 100 mM NaCl, 1 mM MgCl2, 0.05% DDM, 100 µM Tris(2-carboxyethyl)phosphine hydrochloride (TCEP) (Sigma-Aldrich), 10 µM GDP (Sigma-Aldrich). Cleaved heterotrimeric G-protein was further purified by reloading over Ni-sepharose resin. The flow through was collected and purified over a size-exclusion chromatography using a Superdex 200 10/300 column.

Soluble Gβγ subunit for crystallography was expressed and purified from Trichuplusia ni (Hi5) insect cells. Hi5 cells were infected with baculovirus encoding the human β1γ2 subunits with cysteine 68 of γ2 subunit mutated to serine (Gβ1γ2C68S) to remove the geranylgeranylation modification. A decahistidine tag was attached at the amino terminus of the β1 subunit with HRV-3C protease cleavable site. Cultures were harvested 48 h post infection. Cells were lysed in the lysis buffer (10 mM Tris pH 7.4, 5 mM 2-mercaptoethanol, 160 µg/mL benzamidine and 2.5 µg/mL leupeptin). Following to the centrifugation, the supernatant was incubated with Ni-chelating sepharose. The resin was first washed with a high salt buffer (20 mM Hepes pH 7.5, 500 mM NaCl, 20 mM imidazole and 2 mM 2-mercaptoethanol) then a low salt buffer (20 mM Hepes pH 7.5, 100 mM NaCl, 20 mM imidazole and 100 µM TCEP). The protein was eluted with an elution buffer (20 mM Hepes pH 7.5, 100 mM NaCl, 250 mM imidazole and 100 µM TCEP) and dialysed against 20 mM Hepes pH 7.5, 100 mM NaCl and 100 µM TCEP after adding HRC-3C protease to cleave amino-terminal His-tag. Gβ1γ2C68S was further purified by reloading over Ni-sepharose resin. The flow through was collected and purified by a size-exclusion chromatography using a Superdex 200 10/300 column.

The human Gα$_{i1}$ protein for crystallography was expressed in *Escherichia coli* Rosetta2 (DE3) cells (Novagen) with an N-terminal octahistidine-tag and an HRV-3C protease recognition site. The culture was grown at 37 °C in TB medium. When the OD 600 reached 0.6, the protein expression was induced with 0.5 mM IPTG and further grown for 20 h at 24 °C. The cells were harvested by centrifugation, resuspended in the lysis buffer (50 mM Tris-HCl, pH 7.5, 100 mM NaCl, 10 mM imidazole, 0.1 mM PMSF, 10 µM GDP and 5 mM 2-mercaptoethanol). The resuspended cells were disrupted by sonication. Cell lysate was clarified by centrifugation and the supernatant was incubated with Ni-chelating sepharose equilibrated with the lysis buffer. The resin was first washed with the high salt buffer (20 mM Hepes pH 7.5, 500 mM NaCl, 1 mM MgCl$_2$, 20 mM imidazole, 10 µM GDP and 2 mM 2-mercaptoethanol) then the low salt buffer (20 mM Hepes pH 7.5, 100 mM NaCl, 1 mM MgCl$_2$, 20 mM imidazole, 10 µM GDP and 100 µM TCEP). The protein was eluted with the elution buffer (20 mM Hepes pH 7.5, 100 mM NaCl, 1 mM MgCl$_2$, 250 mM imidazole, 10 µM GDP and 100 µM TCEP) and dialysed against 20 mM Hepes pH 7.5, 100 mM NaCl, 1 mM MgCl$_2$, 10 µM GDP and 100 µM TCEP after adding HRC-3C protease to cleave amino-terminal tag. Cleaved Gα$_{i1}$ was further purified by reloading over Ni-sepharose resin. The flow through was collected and purified over a size-exclusion chromatography using a Superdex 200 10/300 column.

GPCR/G-protein complex was prepared essentially in the same way as described previously using agonists lofentanil, iperoxo, BI-167107 for µOR, M$_1$R and M$_2$R, β2AR, respectively[2]. Briefly, receptor was mixed with 1.2–1.5 molar excess G-protein. Following the incubation at room temperature for 1 h, apyrase was added and the reaction mixture was transferred to 4 °C and further incubated for 4 h to overnight. Prior to loading M1 FLAG column, 1% MNG and 0.1% CHS was added. The MNG concentration was progressively lowered during M1 FLAG wash. FLAG eluted protein was further purified by size-exclusion chromatography on a Superdex 200 10/300 column.

**Monoclonal antibody production and characterization.** For the antigen, rhodopsin–G$_{i1}$ complex was stabilized by crosslink using BS-3 (ThermoFisher). Naval Medical Research Institute (NMRI) mice were immunized intraperitoneally with the emulsified antigen. (This study was carried out in strict accordance with the Rules and Regulations for the Protection of Animal Rights (Tierschutzverordnung) of the Swiss Bundesamt für Veterinärwesen. The protocol was ethically approved by the Ethikkommission beider Basel (Permit Number: 237/23523).) Mice with strong ELISA reaction to the antigen were killed and the spleen was removed. Isolated splenocytes were fused with the myeloma cell partner (PAI

mouse myeloma cells, derived from P3-x63-AG8) using polyethylene glycol 1500 (Roche Diagnostics). The fusion mixture was plated into multi-well plates (Thermo Scientific Nunc MicroWell Cell Culture High Flange 96-Well Microplates) and clonal hybridomas were selected by growing in HAT medium supplemented with culture supernatant of mouse macrophages P388. IgG positive clones were screened by ELISA for reactivity against Rhodopsin/$G_{i1}$ complex. Clones that showed a positive reaction in an ELISA assay and by immunoprecipitation were further characterized as monoclonal antibodies or Fab fragments. Initial SEC analysis using rhodopsin/$G_{i1}$ or each component was carried out in 20 mM Hepes pH 7.5, 100 mM NaCl, 2 mM 2-mercaptethanol and 0.01% MNG using Superdex 200 10/200 column.

Coding regions of the heavy-chain (VH-CH1) and light-chain (VL-CL) of mAb16 were cloned into the modified pVL1392 vector where VH-CH1 and VL-CL both attached with GP67 secretion signal sequence were under polyhedron and p10 promoter regulation, respectively. Octahistidine-tag with HRV-3C protease cleavable site was attached to the carboxy-terminus of VH-CH1 for the purification. The single-chain variable fragment of mAb16 (scFv16) was cloned into a modified pVL1392 vector containing a GP67 secretion signal immediately prior to the amino terminus of the scFv16. Octahistidine-tag with HRV-3C protease cleavable site was attached to the carboxy-terminus.

Both Fab16 and scFv16 were expressed in secreted form from Trichuplusia ni Hi5 insect cells using the baculovirus infection method (Expression Systems), and purified by Ni-sepharose chromatography. Supernatant from baculovirus infected cells was pH balanced by addition of Tris pH 7.5. Chelating agents were quenched by addition of 1 mM nickel chloride and 5 mM calcium chloride and incubation with stirring for 1 h at 25 °C. Resulting precipitates were removed by centrifugation and the supernatant was loaded over Ni-sepharose chromatography column. The column was washed with a high salt buffer (20 mM Hepes pH 7.5, 500 mM NaCl and 20 mM imidazole) followed by a low salt buffer (20 mM Hepes pH 7.5, 100 mM NaCl and 20 mM imidazole). The protein was eluted with the elution buffer (20 mM Hepes pH 7.5, 100 mM NaCl and 250 mM imidazole) and the carboxy-terminal octahistidine tag was cleaved by incubation with HRV-3C protease during dialysis against a buffer consisting of 20 mM Hepes pH 7.5 and 100 mM NaCl. Cleaved protein was further purified by reloading over Ni-NTA resin. The flow through was collected and purified over a size-exclusion chromatography using a Superdex 200 16/60 column (GE healthcare). Monomeric fractions were pooled, concentrated and flash frozen in liquid nitrogen until use.

For the binding assay of Fab16 with heterotrimeric G-protein subtypes, Fab16 was first labelled with Alexa Fluor 488 NHS Ester (ThermoFisher Scientific) in 20 mM MES pH 6.5. Free dye was removed by G-50 desalting column (GE healthcare), and the labelled Fab16 was recovered and concentrated. 5–30 µM of G-protein was mixed with 0.4 µM of labelled Fab16, incubated for 1 h and run on SEC on a Superdex 200 10/300 column. Fluorescence signal was recorded with excitation at 488 nm and emission at 512 nm.

**Construction of chimeric G proteins.** $G\alpha_{siN}$ was constructed by substituting the residues 1–38 of $G\alpha_s$ with the residues 1–31 of $G\alpha_{i1}$. $G\alpha_{11iN}$ were constructed by substituting the residues 1–36 of $G\alpha_{11}$ with the residues 1–29 of $G\alpha_{i1}$. These constructs were cloned into pFastBac1 vector and baculovirus were made according to the manufacturer. Scanning chimeras of $G\alpha_{i1}$ as well as $G\alpha_{siN18}$, $G\alpha_{11iN18}$ and $G\alpha_{12iN18}$ are also made in the same way. Primer sequences used to construct the chimeras are provided in the Supplementary Table. The amino-acid sequences of the chimera constructs are provided in the Supplementary Note 2.

**Characterizing resistance to GTPγS.** GTPγS resistance test was performed in the buffer containing 20 mM Hepes pH7.5, 100 mM NaCl, 100 µM TCEP, 0.01% MNG, with each agonist for the complex, lofentanil, iperoxo, BI167107 for µOR/$G_{i1}$, $M_1R/G_{11iN}$ and $M_2R/G_{oA}$, β2AR/$G_{siN}$, respectively. Purified complex with or without Fab16 was incubated with 100 µM GTPγS in the buffer and incubated for 1 h at 24 °C followed by SEC analysis on Superdex 200 10/300 monitoring the protein intrinsic fluorescence with the excitation wavelength at 280 nm and emission wavelength 340 nm.

**Determination of the structure of the $G_{i1}$/scFv16 complex.** For $G_{i1}$/scFv16 crystallization, separately purified and concentrated $G\alpha_{i1}$, Gβ1γ2C68S and scFv16 were mixed at a 1:1:1.2 molar ratio and incubated for 30 min at 24 °C. The resulting $G_{i1}$/scFv16 complex was purified from uncomplexed subunits and free scFv16 by SEC in the buffer containing 20 mM Hepes pH7.5, 100 mM NaCl, 1 mM MgCl2, 10 µM GDP and 100 µM TCEP. Purified $G_{i1}$/scFv16 was incubated with 1 mM aluminium chloride and 50 mM sodium fluoride for 1 h on ice, concentrated to 10–15 mg/mL and crystallized using the hanging drop vapour diffusion method at 20 °C against a reservoir solution containing 10% PEG 8000, 0.1 M Sodium citrate pH 5.0, 1 mM MgCl2, 10 µM GDP, 100 µM TCEP, 1 mM aluminium chloride and 10 mM sodium fluoride. Crystals appeared within a few hours and grew to the full size in 5 days. Crystals were soaked into the reservoir solution supplemented with 25% glycerol as a cryo-protectant and flash frozen in liquid nitrogen. The X-ray data set was collected at the experimental station 12-2 in the Stanford Synchrotron Radiation Lightsource. Diffraction data were integrated by XDS[36], scaled and

merged by AIMLESS[37]. The structure was solved by the molecular replacement in Phaser[38] using the heterotrimeric Gi-protein (1GP2) and scFv fragment (4NKD) as independent search models. Manual model building was performed in Coot[39] and refinement was performed with Phenix refine[40,41]. Ramachandran statistics are favoured 96.6%, allowed 3.4%, outlier 0.0%.

**Nucleotide-binding studies.** For the nucleotide-binding experiment, fluorescence from BODIPY-FL-GTPγS or BODIPY-FL-GDP (ThermoFisher Scientific) was recorded using the Fluorolog spectrophotometer (HORIBA) in the 500 µL quartz cuvette. The fluorophore was exited at 495 nm and emission was detected at 508 nm at 22 °C. The buffer composition is 20 mM HEPES, pH 7.5, 100 mM sodium chloride, 0.01% LMNG, 10 mM magnesium chloride, 100 µM TCEP and 10 µM iperoxo or BI-167107 for GPCR/G-protein complexes, and 20 mM HEPES, pH 7.5, 100 mM sodium chloride, 0.02% DDM, 10 mM magnesium chloride, 100 µM TCEP for Gi1. Kinetics data were collected with 1 µM fluorophore alone for 100 s to establish the baseline fluorescence intensity. Protein was added to 200 nM and rapidly mixed in the fluorescence cuvette. Data points were acquired every second for 600 s. The resulting kinetics spectra were plotted and fit to one phase- or two phase-association function using GraphPad Prism 7.0.

**Data availability.** Data supporting the findings of this manuscript are available from the corresponding authors upon reasonable request. Structure and data set in this work have been deposited in the Protein Data Bank under accession code PDB 6CRK.

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

## Acknowledgements

We thank Jean-Philippe Carralot (F. Hoffmann-La Roche Ltd) for help in antibody generation, Martin Siegrist, Georg Schmid, Bernard Rutten, Doris Zulauf, Stephanie Kueng (Roche Non-Clinical Biorepository) and Ralf Thoma for technical assistance for biomass and cell line generation. Shoji Maeda was supported by the Roche Postdoctoral Fellowship (RPF ID: 113). This work was supported by NIH grant R01GM083118 to B.K. K. B.K.K. is a Chan Zuckerberg Biohub investigator.

## Author contributions

S.M. prepared rhodopsin/Gi1 complex for immunization and selection, performed mAb characterization, Fab and scFv cloning, expression and purification with assistance from A.M., prepared M1R/G11iN, M2R/GoA, β2AR/GsiN, performed Gi1/scFv16 crystallization, structure determination and characterization. A.K. prepared µOR/Gi1. H.M. performed immunization and antibody selection with assist from R.J.P.D. H.H. performed negative stain EM visualization. D.H. prepared β2AR/Gs and provided advice for G-protein experiments. G.S. provided advice on EM analysis and interpretation. G.F.X.S initiated the project. R.J.P.D., and B.K.K. supervised the project.
