## [Peer Review File · Nature Communications]

Reviewers' comments:

Reviewer #1 (Remarks to the Author):

This paper by Maeda et al describes the development of antibodies that can be used to stabilise GPCR G protein complexes for structural studies. Structural studies on activated GPCR complexes are extremely challenging due to the flexibility and instability of these complexes. To date, only structures bound by Gs have been reported in the literature and despite recent advances, complex formation and stabilisation with non-Gs proteins remains a significant challenge. The novel antibody and minimal engineering methods described in this study have the potential to aid in the generation of stable GPCR complexes of receptors coupled to non-Gs proteins that may enable active structures of numerous GPCRs to be solved. Indeed, this group have already used this method to solve a structure of a receptor bound to Gi that is currently under review. While this novel methodology will provide an extremely useful tool for the GPCR structural biology community, I have a few concerns that need to be addressed prior to recommending publication in Nature Communications.

The authors show convincingly that G alpha N chimeras can be generated that allow the antibody to recognise Gs and Gq subclasses. The chimeras used to do this replace the majority of the alpha N subunit, not just the region that is recognised by the antibody. The current Gs bound structures reveal that the C-terminus of the alphaN helix resides close to the intracellular loops of the receptors to which they are bound, particularly intracellular loop 2. The chimeras used in this paper, while functionally still selective to their respective signalling pathways, may form different interactions with the receptors due to the full replacement of the alpha N helix. Indeed, many GPCRs can bind multiple G proteins, with Gi being the most promiscuous coupler for most GPCRs. For these tools to be most applicable for the community, it would be important to understand the minimal requirement for engineering non-Gi/o G proteins that still enable the binding of the antibody and formation of stable receptor:GPCR complexes. For example, the authors nicely show in supplementary figure 1 that minimal substitutions within the alpha N helix of G11 are still able to express and form a heterotrimer. Just replacing the antibody interacting amino acids (f) or a more limited region of the N-terminus of the alphaN (such as chimera a in supplementary figure 1) would only alter the G protein in a region that would be predicted to be far away from the receptor and not alter receptor:Gprotein interactions. Can either of these chimeras shown still interact with the antibody? And can these form stable complexes with the receptor?

Additionally, the authors mention G12/13 family, but do not show any data that this tool is applicable for stabilising complexes of this subfamily. For mAb16 to really be a universal tool, there is a requirement for it to bind G proteins from all 4 major classes. Is engineering of G12/13 proteins possible that enable this antibody to bind and can these tools be used to stabilise a receptor:G12/13 complex?

Minor comments

The authors show an EM grid of the purified the M1R/G11iN/scFv16 complex, but no 2D class averages of their complex. I would like to see these.

What is the SEC trace showing in the top left panel of figure 1C? Is this just Galpha or is it the heterotrimer? If this is just G alpha, and the antibodies (with the exception of mAb16) only recognise the Gbeta/gamma, why is there a shift in the SEC trace with all antibodies?

Reviewer #2 (Remarks to the Author):

This is an interesting and highly valuable paper which characterizes a new monoclonal antibody and fragment which will be useful to stabilize GPCR-G protein complexes for structural studies.

Apparently the authors have already used this reagent to obtain a structure of the mu-opioid receptor in complex with heterotrimeric Gi and in the paper they demonstrate its utility for other GPCRs and engineered G proteins.

The antibody has some specificity for Gi/o family of G protein alpha subunits and does not interact with native Gs of Gq alphas; G12 was not tested. With some modest engineering they are able to show stabilization of Gs-bAR and G11-M2 complexes.

Reading the title and abstract a naive reader would assume a 'universal' reagent was achieved, although reading the paper a bit more carefully it is clear that for Gs, Gq and G12 families some engineering of the N-terminus of the alpha subunits are needed (and perhaps this will need to be ultimately optimized for every GPCR-G protein complex attempted).

Thus the reagent is not 'universal' in the most commonly used sense of the word and I would strongly recommend deleting this from the title and abstract.

The last sentence of the abstract also needs to be modified to make clear that engineering of alpha subunits will be required unless this is a Gi/o complex.

As well the full sequence of the antibody and the fragment need to be provided as a supplement to the main document (even though this information should be available in the PDB file) as if this is as useful as it appears many labs will want the full sequence immediately upon publication.

I would also recommend that the authors consider making the antibody cDNA as well as the fragment available via a repository (eg ADDGENE) with the standard MTAs for non-profits.

Reviewer #3 (Remarks to the Author):

The paper by Maeda et al. is well written and provides data on an antibody, mAb16, and its Fab and scFV antibody fragment derivatives that stabilize the interface between Gai1 and Gβ2 and is proposed as a useful tool for structural studies on G protein-coupled receptors (GPCRs). Relative small modifications to other Gα subunits allow this reagent to be transferred to use on other members of this family. This reagent is likely to be a very useful one for GPCR structural studies.

This antibody appears to have been developed by Roche Parma. There is no sequence data or indication the hybridoma or derived scFV construct have been, or will be, deposited in a publicly available repository. Given the potential intellectual property claims surrounding this tool I would like to see a clear statement on how this tool will be shared with the wider scientific community, particularly given this is a methods paper.

For the general reader the SEC traces presented in figures 1, 3 and 5 are difficult to interpret. These figures would benefit greatly from having the most important fractions isolated and shown as western blots to confirm the identity of the co-migrating species from the SEC. In figure 1a, a western blot showing the fraction peak at ~7mL and 9mL and probed for the mAb, Rhodopsin and Gai would help the general reader understand what is being assessed. In figure 1c, a western blot for the 7mL and 11mL fractions from the Gai1βγ +/- mAb16 should be provided. Similarly, the SEC profiles in figure 3 and 5 are difficult for a generalist reader to follow, if western blots could be provided for relevant fractions for figure 3c, this would provide the reader with confidence that the claims about the peak composition are reasonable.

With the exception of the difficulty in interpretation of the SEC traces the figures are generally well presented, however there is no indication as to how reproducible the data is. Are the SEC traces in figure 1, 3 and 5 representative of more than one experiment or were these experiments only performed once? A clear indication as to reproducibility needs to be made. The nucleotide

association curves shown in figure 4 have no error bars and there is no indication in the text or figure legend how many independent experiments were used to derive this data, this needs to be provided. The parameters for association rate derived from the data presented in figure 4 are provided in table 2 and are quoted to 4 significant figures with no error, how is this justified given it appears that the data might be derived from a single experiment?

The claim of universality of this reagent is an over-reach. There are 5 different G β isoforms and although G β 1 has been shown to be able to partner with most G α isoforms it is not the preferred partner for all G α isoforms. Although the authors make a very good case for binding a variety of G α isoforms, representing all subfamilies, not every G α is tested and thus any claim for universality is over-stated. I would suggest using the term "broadly" (or something similar) to replace "universally".

Minor comments:

Page 3, line 51, delete of: should read "... human genome, comprising around 800 ..."

Page 6, line 150 complementarity determining region is often hyphenated: complementarity-determining region

Page 10, line 184 reads "structures are shown as cylinder (helix) and allow (strand)", replace allow with arrow, should read: "structures are shown as cylinder (helix) and arrow (strand)"

Figure 4, B, key to GDP binding curve at bottom is mis-labelled as "Gi1 Fa16", should be: "Gi1 Fab16"

Figure 5, in the legend the 2 SEC plots are indicated as b and c and the negative stain EM as d but in the figure both SEC plots are b and the EM is c. The EM has no scale bar.

Supplementary Figure 1, it would be helpful to have an arrow indicating the scFV on the coomassie stained gel and it would be useful to indicate that this is, indeed a coomassie stained gel.

Reviewer #4 (Remarks to the Author):

>The manuscript describes the identification of a mAb that will greatly facilitate cryo-EM studies of GPCR/G protein complexes. A murine monoclonal antibody "mAb16" that binds to the heterotrimeric form of rhodopsin/Gi1 complex was isolated. The antibody in scFv format was co-crystallised with the Gi1 complex and the structure was solved. The epitope was identified to lie within the α N helix of G α i1 and G β , and is within a sequence that is highly conserved among Gi/o proteins. Indeed mAb16 was shown to bind a panel of 5 other Gi/o proteins but not a Gs protein. Furthermore the epitope was verified by a generating panel of α N chimeras with other G-protein α -subunits. mAb16 also inhibited GTP γ S induced dissociation of the G-protein complex.

>The mAb that has been isolated is of significant value in aiding elucidation of the 3D structure of important and high-value Gi/o GPCR drug targets.

> The data are very convincing. The experiments were carefully planned and the manuscript is well written.

> Minor points:

Line 177 – consider re-wording to "...binds to five different Gi/o type G-proteins but not a Gs..."

Line 196 – consider re-wording to "...to a panel of Gi/o family members..."

Although you clarify this nicely in the discussion paragraph 2 from line 327.

Figure 5 – “c” mis-assigned, “d” missing

Line 184 - allow > arrow

We thank the reviewers for their constructive comments. The reviewers' comments are in blue text and our responses are in black text.

Reviewer #1 (Remarks to the Author):

This paper by Maeda et al describes the development of antibodies that can be used to stabilise GPCR G protein complexes for structural studies. Structural studies on activated GPCR complexes are extremely challenging due to the flexibility and instability of these complexes. To date, only structures bound by Gs have been reported in the literature and despite recent advances, complex formation and stabilisation with non-Gs proteins remains a significant challenge. The novel antibody and minimal engineering methods described in this study have the potential to aid in the generation of stable GPCR complexes of receptors coupled to non-Gs proteins that may enable active structures of numerous GPCRs to be solved. Indeed, this group have already used this method to solve a structure of a receptor bound to Gi that is currently under review. While this novel methodology will provide an extremely useful tool for the GPCR structural biology community, I have a few concerns that need to be addressed prior to recommending publication in Nature Communications.

The authors show convincingly that G alpha N chimeras can be generated that allow the antibody to recognise Gs and Gq subclasses. The chimeras used to do this replace the majority of the alpha N subunit, not just the region that is recognised by the antibody. The current Gs bound structures reveal that the C-terminus of the alphaN helix resides close to the intracellular loops of the receptors to which they are bound, particularly intracellular loop 2. The chimeras used in this paper, while functionally still selective to their respective signalling pathways, may form different interactions with the receptors due to the full replacement of the alpha N helix. Indeed, many GPCRs can bind multiple G proteins, with Gi being the most promiscuous coupler for most GPCRs. For these tools to be most applicable for the community, it would be important to understand the minimal requirement for engineering non-Gi/o G proteins that still enable the binding of the antibody and formation of stable receptor:GPCR complexes. For example, the authors nicely show in supplementary figure 1 that minimal substitutions within the alpha N helix of G11 are still able to express and form a heterotrimer. Just replacing the antibody interacting amino acids (f) or a more limited region of the N-terminus of the alphaN (such as chimera a in supplementary figure 1) would only alter the G protein in a region that would be predicted to be far away from the receptor and not alter receptor:Gprotein interactions. Can either of these chimeras shown still interact with the antibody? And can these form stable complexes with the receptor?

Additionally, the authors mention G12/13 family, but do not show any data that this tool is applicable for stabilising complexes of this subfamily. For mAb16 to really be a universal tool, there is a requirement for it to bind G proteins from all 4 major classes. Is engineering of G12/13 proteins possible that enable this antibody to bind and can these tools be used to stabilise a receptor:G12/13 complex?

As suggested by the reviewer, we have made "minimal" chimeric constructs for Gs, G11, and G12 where the initial 18 amino-acid residues of Gi1 are substituted into the equivalent region of Gs, G11, and G12. For Gs and G11 chimera, we tested formation of stable GPCR/G-protein complex and binding of Fab16. For G12 we tested the binding of Fab16 to the engineered G12iN. We've provided these data in the Supplementary Figure 1.

Minor comments

The authors show an EM grid of the purified the M1R/G11iN/scFv16 complex, but no 2D class averages of their complex. I would like to see these.

We performed the negative-stain EM (Fig. 5d) to show that we had uniform complexes, but did not take enough images for generating 2D class averages.

What is the SEC trace showing in the top left panel of figure 1C? Is this just Galpha or is it the heterotrimer? If this is just G alpha, and the antibodies (with the exception of mAb16) only recognise the Gbeta/gamma, why is there a shift in the SEC trace with all antibodies?

The top panel of Figure 1C shows no binding of mAbs to G alpha subunit. The peaks 8.4-9mL correspond to the ones from each mAb and the peak intensity from G alpha at 11mL is unchanged throughout the experiments, indicating that there is no binding between mAb and G alpha. However, we agree that it would be hard to interpret the SEC profiles as they were. We've added more detailed description in the figure panels and the Figure legend.

Reviewer #2 (Remarks to the Author):

This is an interesting and highly valuable paper which characterizes a new monoclonal antibody and fragment which will be useful to stabilize GPCR-G protein complexes for structural studies. Apparently the authors have already used this reagent to obtain a structure of the mu-opioid receptor in complex with heterotrimeric Gi and in the paper they demonstrate its utility for other GPCRs and engineered G proteins. The antibody has some specificity for Gi/o family of G protein alpha subunits and does not interact with native Gs or Gq alphas; G12 was not tested. With some modest engineering they are able to show stabilization of Gs-bAR and G11-M2 complexes.

Reading the title and abstract a naive reader would assume a 'universal' reagent was achieved, although reading the paper a bit more carefully it is clear that for Gs, Gq and G12 families some engineering of the N-terminus of the alpha subunits are needed (and perhaps this will need to be ultimately optimized for every GPCR-G protein complex attempted).

Thus the reagent is not 'universal' in the most commonly used sense of the word and I would strongly recommend deleting this from the title and abstract.

The last sentence of the abstract also needs to be modified to make clear that engineering of alpha subunits will be required unless this is a Gi/o complex.

We believe the sentences from line 44 to 46 explain the need of modification in non- Gi/o proteins. We understand the concern raised by the reviewer on the word 'universal'. We have deleted the word "universal" from the title (line 1), and replaced with "broadly applicable" in the summary and the discussion (line 47, line 301).

As well the full sequence of the antibody and the fragment need to be provided as a supplement to the main document (even though this information should be available in the PDB file) as if this is as useful as it appears many labs will want the full sequence immediately upon publication.

We have provided the full sequence of mAb16 in the Supplementary Material 1. We've also mentioned the availability of materials in line 324-326.

I would also recommend that the authors consider making the antibody cDNA as well as the fragment available via a repository (eg ADDGENE) with the standard MTAs for non-profits.

Hybridoma cells, antibodies, plasmids for the antibody fragments are publicly available from Roche by making a standard MTA. We have added a sentence on the availability of materials in line 324-326.

Reviewer #3 (Remarks to the Author):

The paper by Maeda et al. is well written and provides data on an antibody, mAb16, and its Fab and scFV antibody fragment derivatives that stabilize the interface between G α 1 and G β 2 and is proposed as a useful tool for structural studies on G protein-coupled receptors (GPCRs). Relative small modifications to other G α subunits allow this reagent to be transferred to use on other members of this family. This reagent is likely to be a very useful one for GPCR structural studies.

This antibody appears to have been developed by Roche Parma. There is no sequence data or indication the hybridoma or derived scFV construct have been, or will be, deposited in a publicly available repository.

Given the potential intellectual property claims surrounding this tool I would like to see a clear statement on how this tool will be shared with the wider scientific community, particularly given this is a methods paper.

We have added a sentence that the materials are publicly available from Roche by making a standard MTA (line 324-326). Also we have provided a full sequence of mAb16 in the Supplementary Material 1.

For the general reader the SEC traces presented in figures 1, 3 and 5 are difficult to interpret. These figures would benefit greatly from having the most important fractions isolated and shown as western blots to confirm the identity of the co-migrating species from the SEC. In figure 1a, a western blot showing the fraction peak at ~7mL and 9mL and probed for the mAb, Rhodopsin and Gai would help the general reader understand what is being assessed. In figure 1c, a western blot for the 7mL and 11mL fractions from the Gai1 β +/- mAb16 should be provided. Similarly, the SEC profiles in figure 3 and 5 are difficult for a generalist reader to follow, if western blots could be provided for relevant fractions for figure 3c, this would provide the reader with confidence that the claims about the peak composition are reasonable.

Unfortunately, we did not retain the fractions from the analytical SEC experiments for western blot analysis. However, to address the reviewer's concern, we have modified the figures to more clearly label the SEC peak positions for individual proteins and complexes. To demonstrate the composition of the complexes with FAB16, we include coomassie-stained gels of peak fractions for complexes formed with engineered Gs (GsiN18), G11 (G11iN18) and G12 (G12iN18) in Supplementary Figure1.

With the exception of the difficulty in interpretation of the SEC traces the figures are generally well presented, however there is no indication as to how reproducible the data is. Are the SEC traces in figure 1, 3 and 5 representative of more than one experiment or were these experiments only performed once? A clear indication as to reproducibility needs to be made. The nucleotide association curves shown in figure 4 have no error bars and there is no indication in the text or figure legend how many independent experiments were used to derive this data, this needs to be provided. The parameters for association rate derived from the data presented in figure 4 are provided in table 2 and are quoted to 4 significant figures with no error, how is this justified given it appears that the data might be derived from a single experiment? The SEC experiments in Fig. 1 were performed only once, but these were initial studies to identify a mAb that bound to an α/β interface. Subsequent experiments, including the crystal structure of the Gi1/Fab16 complex confirmed the results of these SEC experiments. All other SEC experiments were performed at least twice and the representative results are presented in the figures.

We've updated the nucleotide binding experiments in Figure 4 with one performed in triplicate. The error bars are sometimes very small and might be difficult to see. We have corrected the number of significant figures in Table 2. The number of significant figures is based on the standard error of the measurement.

The claim of universality of this reagent is an over-reach. There are 5 different G β isoforms and although G β 1 has been shown to be able to partner with most G α isoforms it is not the preferred partner for all G α isoforms. Although the authors make a very good case for binding a variety of G α isoforms, representing all subfamilies, not every G α is tested and thus any claim for universality is over-stated. I would suggest using the term "broadly" (or something similar) to replace "universally".

We agree that "universal" might be an overstatement from our experiment that does not test every single G-protein isoform but representative members from all subfamilies. We removed the word "universal" from the title (line 1), and replaced it with "broadly applicable" in the main text (line 47, line 301). We've provided a sequence alignment of 5 different G β isoforms in the supplementary figure2 with mAb16 binding site highlighted. This alignment indicates high sequence identity/similarity in the mAb16 binding site among G β isoforms with exception of G β 5 isoform. We added sentences regarding this alignment in the main text (line 226-229).

Minor comments:

Page 3, line 51, delete of: should read "... human genome, comprising around 800 ..."

Line 51, Corrected as "comprising around 800 members"

Page 6, line 150 complementarity determining region is often hyphenated: complementarity-determining region

Line 143, corrected as "complementarity-determining region"

Page 10, line 184 reads "structures are shown as cylinder (helix) and allow (strand)", replace allow with arrow, should read: "structures are shown as cylinder (helix) and arrow (strand)"

Line 644, corrected as "structures are shown as cylinder (helix) and arrow (strand)"

Figure 4, B, key to GDP binding curve at bottom is mis-labelled as "Gi1 Fa16", should be: "Gi1 Fab16"

Corrected in the updated Figure 4.

Figure 5, in the legend the 2 SEC plots are indicated as b and c and the negative stain EM as d but in the

figure both SEC plots are b and the EM is c. The EM has no scale bar.

The figure labeling was corrected. Our EM is not equipped with a scale bar but based on magnification, the particles are in the appropriate size of GPCR/G-protein/scFv16 complex.

Supplementary Figure 1, it would be helpful to have an arrow indicating the scFv on the coomassie stained gel and it would be useful to indicate that this is, indeed a coomassie stained gel.

We provided additional data on the previous supplementary Figure1, The presence of scFv16 co-eluting with the GPCR/G-protein complexes are shown in the coomassie stained SDS-PAGE with the corresponding SEC profiles.

Reviewer #4 (Remarks to the Author):

>The manuscript describes the identification of a mAb that will greatly facilitate cryo-EM studies of GPCR/G protein complexes. A murine monoclonal antibody "mAb16" that binds to the heterotrimeric form of rhodopsin/Gi1 complex was isolated. The antibody in scFv format was co-crystallised with the Gi1 complex and the structure was solved. The epitope was identified to lie within the α N helix of G α i1 and G β , and is within a sequence that is highly conserved among Gi/o proteins. Indeed mAb16 was shown to bind a panel of 5 other Gi/o proteins but not a Gs protein. Furthermore the epitope was verified by a generating panel of α N chimeras with other G-protein α -subunits. mAb16 also inhibited GTP γ S induced dissociation of the G-protein complex.

>The mAb that has been isolated is of significant value in aiding elucidation of the 3D structure of important and high-value Gi/o GPCR drug targets.

> The data are very convincing. The experiments were carefully planned and the manuscript is well written.

> Minor points:

Line 177 – consider re-wording to "...binds to five different Gi/o type G-proteins but not a Gs..."

Line 196 – consider re-wording to "...to a panel of Gi/o family members..."

Although you clarify this nicely in the discussion paragraph 2 from line 327.

Figure 5 – "c" mis-assigned, "d" missing

Line 184 - allow > arrow

We made the suggested changes.

Line 152 – “bind to all Gi/o type G-proteins” to “bind to five different Gi/o type G-proteins”

Line159 – “bind to all Gi/o family members” to “bind to a panel of Gi/o family members”

Line 644 in the Figure Legend, corrected as “structures are shown as cylinder (helix) and arrow (strand)”

REVIEWERS' COMMENTS:

Reviewer #1 (Remarks to the Author):

The revised version of this manuscript has been improved with additional experiments included. All of my concerns have been addressed.

Reviewer #3 (Remarks to the Author):

Maeda et al. have satisfactorily addressed my comments on their original submission. I congratulate them on a well written and presented methods paper that should provide an enabling resource for future structural studies on GPCR-G protein complexes.